# Influenza Vaccination Coverage among People with Self-Reported Cardiovascular Diseases—Findings from the Hungarian Implementation of the European Health Interview Survey

**DOI:** 10.3390/vaccines12040360

**Published:** 2024-03-27

**Authors:** Gergő József Szőllősi, Jenifer Pataki, Anett Virágh, Gábor Bányai, Klára Boruzs, Klára Bíró, Viktor Dombrádi

**Affiliations:** 1Coordination Center for Research in Social Sciences, Faculty of Economics and Business, University of Debrecen, 4032 Debrecen, Hungary; 2Faculty of Health Sciences, University of Debrecen, 4032 Debrecen, Hungary; pataki.jenifer@etk.unideb.hu (J.P.); viragh.anett7@gmail.com (A.V.); 3Doctoral School of Health Sciences, University of Debrecen, 4032 Debrecen, Hungary; 4Institute of Health Economics and Management, Faculty of Economics and Business, University of Debrecen, 4032 Debrecen, Hungary; banyai.gabor@econ.unideb.hu (G.B.); boruzs.klara@econ.unideb.hu (K.B.); biro.klara@econ.unideb.hu (K.B.); 5Patient Safety Department, Health Services Management Training Centre, Faculty of Health and Public Administration, Semmelweis University, 1085 Budapest, Hungary; dombradi.viktor@emk.semmelweis.hu

**Keywords:** cardiovascular disease, epidemiology, influenza, vaccination coverage

## Abstract

Worldwide, cardiovascular diseases are the leading cause of mortality. This has significant implications for public health. Influenza, a common infectious disease, poses an increased risk for individuals with chronic conditions, such as cardiovascular diseases. However, little is known about influenza vaccination coverage in this group. This study utilized data from the Hungarian implementation of the European Health Interview Survey to assess influenza vaccination coverage and its determinants among cardiovascular respondents from 2009 to 2019. The findings reveal a downward trend in the vaccination rates over the years (from 24% to 21%), despite the availability of free vaccination in Hungary for this high-risk population. The main factors influencing low influenza vaccine uptake were identified, as follows: young age, a lower level of education, good self-perceived health status, smoking, a lower frequency of medical visits, and not suffering from respiratory diseases. Addressing these disparities necessitates targeted vaccination strategies supported by enhanced education, better access to healthcare services, and the promotion of preventive healthcare measures. Improving vaccination coverage among patients with cardiovascular diseases is imperative for reducing influenza-related morbidity and mortality. This highlights the importance of comprehensive public health interventions and healthcare provider engagement in promoting vaccination among groups at increased risk.

## 1. Introduction

Cardiovascular diseases are considered the leading cause of death worldwide. Approximately 17.9 million people lose their lives to this set of diseases annually, accounting for 32% of fatalities worldwide [1]. This group of diseases includes disorders of the heart and blood vessels, such as coronary heart disease, stroke, hypertension, heart failure, rheumatic heart disease, congenital heart disease, peripheral artery disease, and other cardiovascular conditions, which places cardiovascular diseases at the top of public health priorities [2]. In terms of transmission regarding infectious diseases, seasonal influenza spreads easily, as it is rapidly transmitted in crowded areas, including schools and nursing homes. All age groups can be affected, but some groups are more at risk than others [3]. However, morbidity and mortality due to influenza are relatively high, and, in most cases, patients are hesitant to be administered the influenza vaccine each year [4]. This may occur due to a lack of trust in the vaccine or the health care personnel, due to vaccine reactions, or due to the inconvenience of the vaccine [5,6]. People with chronic medical conditions, such as cardiovascular diseases, are at greater risk of severe disease or complications when infected [7]. Influenza and chronic diseases are the subject of several studies, and numerous studies explore the relationship between them [8,9,10,11,12,13]. Influenza, commonly known as viral flu, can cause severe respiratory illness and can be particularly dangerous for those already dealing with chronic conditions. According to the Centers for Disease Control and Prevention (CDC), chronic diseases—such as diabetes, cardiovascular diseases, respiratory disorders, or immune system issues—significantly increase the risk of severe consequences from influenza. Infection caused by influenza can exacerbate these chronic conditions, potentially leading to life-threatening situations [14]. Due to the continuous evolution of influenza viruses, the most effective protection is achieved through annual vaccination. Numerous studies have demonstrated that vaccination not only reduces the severity of the disease but also decreases the likelihood of requiring intensive care in cases of influenza-related complications [8,9,11,12]. Additionally, vaccination lowers hospitalization rates for high-risk patients [9]. According to data from the European Centre for Disease Prevention and Control (ECDC), the vaccination coverage rates in EU/EEA member states are inadequate, particularly among those at greater risk [15]. Increasing vaccination coverage is a key public health priority because vaccination coverage of individuals with cardiovascular and other chronic diseases is an important indicator of the country’s public health status and preparedness. Even though this is an important area, not only in the field of public health in Hungary but also worldwide, there is a knowledge gap about exactly what factors contribute to vaccine acceptance, especially among those with cardiovascular diseases.

Taking all of this into consideration, the main objective of this study was to assess influenza vaccination coverage and its influencing factors among individuals with self-reported cardiovascular disease based on the Hungarian implementation of the European Health Interview Survey (EHIS) studies of 2009, 2014, and 2019. Nationwide representative cross-sectional studies such as EHIS were designed to provide core health information regarding the population of the countries involved, and data regarding vaccination can be considered valid due to the strict regulations applied during data collection. Therefore, the investigation of factors related to vaccination coverage for the country is considered as executable. This approach is supported by the fact that other countries have also conducted similar studies to determine the factors that might influence vaccination coverage [16,17,18,19,20,21,22].

## 2. Materials and Methods

### 2.1. Database

The data were obtained from the 2009, 2014, and 2019 Hungarian implementations of the cross-sectional EHIS, which was conducted in order to establish reliable health indicators regarding health status, healthcare, and health determinants with background variables, such as demography and socio-economic characteristics, in EU Member States and Serbia. The survey collects information on the population’s health status, lifestyle characteristics, limitations of self-care, physical activity, nutrition, health risk behaviors, healthcare utilization, and satisfaction. The multistage stratified sampling method was developed by the Hungarian Central Statistical Office, and the addresses of the potential respondents were provided by the Ministry of Interior. Trained personnel of the Hungarian Central Statistical Office conducted data collection on representative samples using a standardized questionnaire under the supervision of Eurostat. While the databases themselves are not publicly available, they can be requested from the Hungarian Central Statistical Office, which supervised the data collection and primary analysis. During the studies, stratified probability samples were used for creating reliable indicators of the Hungarian adult population living in private households. Because the data collection methodology remained consistent across all three surveys, the three datasets and their respective sets of variables were treated as equivalent. Consequently, the data from all three surveys were merged into a single dataset [23].

### 2.2. Data

In this study, the primary outcome was based on self-reporting, as the main dependent variable was influenza vaccine uptake within a year prior to the EHIS study. The answer options were dichotomized; thus, it was considered positive when subjects had received their last influenza vaccination within a year, and it was considered negative when subjects had received their last influenza vaccination more than a year ago. This was essential because influenza varies from year to year. For this reason, we examined whether the respondents of this study were considered protected in the given years. No data were available on the indication for influenza vaccination. The independent variables were sex (male/female), age group (18–64/65–x), and education level (primary/secondary/tertiary). Furthermore, the following variables were added to the independent variables: marital status, perceived financial situation, self-rated health, body mass index categories, smoking status, and encounters with a general practitioner (GP) or specialist. For marital status, we distinguished between individuals living alone and those in a marital/partner relationship. We categorized perceived financial situations into average/good and poor financial conditions. Based on self-rated health status, we classified respondents into good and poor self-rated health. For body mass index (BMI) categories, we determined whether the respondents were considered overweight or obese. For smoking status, respondents were divided into smokers (yes) and non-smokers (no). Regarding encounters with a general practitioner, we distinguished between those who had seen their GP within the recent year and those who had not seen their GP for more than a year. Data on respiratory diseases (yes/no) were assessed on an aggregated basis, aggregating if the respondent had chronic bronchitis, emphysema, or asthma. All variables were self-reported. In this study, the analysis focused on individuals suffering from cardiovascular diseases based on self-reporting.

### 2.3. Statistical Methods

A multivariate logistic regression model was executed using Stata statistical software (13.0-s version, Stata Corp., College Station, TX, USA). For data description, univariate descriptive statistics were used, and we employed chi-squared tests. The data were presented with raw case numbers, as well as proportions. The results were evaluated through analyses with odds ratios adjusted for all potential confounding factors—with their corresponding *p*-values—included in the analysis. All the multiple analyses were adjusted for the respondent’s place of residence to adjust the analysis for potential territorial effects. The results were considered significant if the *p*-values were less than 0.05.

## 3. Results

### 3.1. Descriptive and Chi-Squared Results

The initial sample size was 16,480. The prevalence of self-reported cardiovascular diseases was 32% in 2009 and 33% in 2014; it significantly (*p* < 0.001) increased to 35% in 2019. Subsequently, selection criteria were applied, which meant that the study sample was reduced to patients with cardiovascular diseases only. This data cleaning involved the exclusion of respondents for whom data were not available regarding any of the variables of potential interest, which is why the total sample size containing data regarding patients with cardiovascular diseases was reduced to 1997 respondents from the 2009 dataset, 2096 from the 2014 dataset, and 2179 participants from the 2019 dataset; therefore, the merged sample consisted of 6272 individuals’ data. Over the years, a borderline-significant (*p* = 0.065) difference in vaccination rates has been observed. In 2009, among self-reported patients with cardiovascular diseases, 479 individuals (24%) received the influenza vaccination. In 2014, 467 individuals (22%) received it, and, in 2019, 457 individuals (21%) with self-reported cardiovascular issues were vaccinated. The elderly had significantly (*p* < 0.001) higher vaccination rates during the years compared with adults aged 18–64, since 38% (n = 305) of the elderly subjects received vaccination in 2009, 34% (n = 305) received it in 2014, and 29% (n = 347) received vaccination in 2019 (Table 1). Meanwhile, the percentage of adults aged 18–64 who received influenza vaccination within a year was 15% (n = 174) in 2009, 14% (n = 162) in 2014, and 11% (n = 110) in 2019. A significant (*p* < 0.001) difference was observed in the merged sample, since the vaccination coverage within a year was higher among elderly subjects (n = 957, 33%) compared with adults aged 18–64 (n = 446, 13%). No significant difference was observed in either the year-by-year analysis or the merged analysis between influenza vaccination status and sex. Female respondents had higher vaccination rates; however, the difference was not significant. Educational level proved to be a significant factor in relation to influenza vaccination coverage, but no clear trend was observed during the studies. In 2009, the highest vaccination coverage was observed among those with a primary education level (n = 196, 27%, *p* < 0.001). However, in 2014 (n = 105, 25%, *p* = 0.007) and 2019 (n = 112, 29%, *p* < 0.001), the highest vaccination coverage was observed among respondents with a tertiary education level. No statistically proven difference was found in the uptake of the influenza vaccine between respondents living in rural or urban areas. Furthermore, marital status was not found to be a clear significant determinant of influenza vaccination among cardiovascular respondents. Self-rated income proved to be a significant influencing factor in the year 2019, since respondents with low income had lower vaccination coverage (n = 177, 18%) compared with respondents with good income (n = 280, 23%). Self-perceived health status was significantly (*p* < 0.05) associated with vaccination in all three years and in the merged sample. Respondents with poor self-perceived health status had significantly higher vaccination rates within a year (2009: n = 171, 28%; 2014: n = 140, 27%; 2019: n = 137, 28%; merged sample: n = 448, 28%) compared with participants with good self-perceived health status (2009: n = 308, 22%; 2014: n = 327, 21%; 2019: n = 320, 19%; merged sample: n = 955, 21%). Prevalence of obesity or overweight status—created by calculating BMI and dichotomizing the variable—was not found to be an influencing factor regarding vaccination in any year (*p* ≥ 0.05). Cardiovascular respondents who were smokers were significantly (*p* < 0.001) less likely to be vaccinated. Furthermore, influenza vaccination coverage was reasonably higher among non-smokers in 2009 (n = 411, 27%), 2014 (n = 418, 25%), 2019 (n = 396, 23%), and in the merged sample (n = 1225, 25%) compared with smokers (2009: n = 68, 14%; 2014: n = 49, 11%; 2019: n = 61, 13%; merged sample: n = 178, 13%). The exact opposite association was observed for self-reported respiratory diseases among cardiovascular respondents because people who had respiratory diseases (2009: n = 96, 30%; 2014: n = 85, 31%; 2019: n = 80, 28%; merged sample: n = 261, 29%) were significantly (*p* < 0.05) more likely to accept the influenza vaccine than those who did not have such diseases (2009: n = 383, 23%; 2014: n = 382, 21%; 2019: n = 377, 20%; merged sample: n = 1142, 21%). The subjects’ last meetings with a doctor within a year proved to be an associating factor for influenza vaccine coverage and acceptance. Influenza vaccination rates were considered higher among people who made more frequent visits to their general practitioners or specialists. This association was significantly (*p* < 0.05) expressed in all three years of this study, as well as in the merged sample. Respondents from 2009 who had visited their doctor or specialist within a year had 19% higher vaccination coverage (n = 474, 25%) compared with people who visited their doctor more than a year ago (n = 5, 6%). The same trend was observed in all the other years. Cardiovascular respondents of 2014 who had visited their doctor within a year had a higher (n = 458, 23%) vaccination rate compared with those who did not meet with their doctors (n = 9, 10%) regularly. In 2019, patients with frequent visits had a 21% (n = 446) vaccination rate compared with individuals with infrequent visits, whose vaccination rate was 11% (n = 11). The same trend was observed in the merged and matched sample.

### 3.2. Results of the Multiple Logistic Regression Analysis

Age was a significant influencing factor of influenza vaccine acceptance and vaccine coverage among cardiovascular respondents: in 2009, elderly subjects had 3.58 times higher odds (AOR = 3.58 [2.80–4.57]) of receiving a flu shot; in 2014, they had 3.08 times higher odds (AOR = 3.08 [2.43–3.89]) of receiving a flu shot; in 2019, respondents aged 65 or above had 3.17 higher odds (AOR = 3.17 [2.47–4.07]) of receiving immunization against influenza compared to younger respondents (Table 2). The odds of the elderly subjects being properly immunized against influenza virus was 3.22 (AOR = 3.22 [2.80–3.70]) higher compared to adults aged 18–64. Sex was not associated with influenza vaccination among the participants. A higher education level was associated with increased odds of being immunized against influenza: respondents from 2014 with a tertiary education level had 2.21 higher odds (AOR = 2.21 [1.29–3.78]) of having received vaccination compared to respondents with a secondary education level. This association was observed in 2019, where individuals with a tertiary level of education had 74% higher odds (AOR = 1.74 [1.21–2.52]) of being properly immunized compared to those with a secondary level of education. In the merged sample, respondents with tertiary education had 57% higher odds (AOR = 1.57 [1.25–1.97]) compared to participants with secondary education. Type of residence was not statistically significant regarding influenza vaccination. Being married or having a partner showed no significant relationship with influenza vaccination. Poor self-rated income had a negative effect on vaccination; however, the association between income and influenza vaccination was not significant in any of the years studied. Self-perceived health status had a positive effect on influenza vaccination: cardiovascular respondents with poor perceived health status had 30% increased odds (AOR = 1.30 [1.01–1.67]) in 2014, 60% increased odds (AOR = 1.60 [1.24–2.07]) in 2019, and 31% increased odds (AOR = 1.31 [1.13–1.51]) for being vaccinated in the merged sample compared with respondents with good self-perceived health status. Being overweight or obese had no significant relationship with vaccine acceptance in this study. Non-smokers had significantly higher odds of being properly immunized against influenza: non-smokers in 2009 had 53% higher odds (AOR = 1.53 [1.13–2.09]) of receiving a flu shot; in 2014, they had 96% higher odds (AOR = 1.96 [1.39–2.75]); and, in 2019, they had 41% higher odds (AOR = 1.41 [1.03–1.92]) of having an influenza vaccine compared to smokers. In the merged sample, the same trend was found: non-smokers had 60% higher odds (AOR = 1.60 [1.34–1.93]) of having received a flu shot compared to smokers. Those participants who were suffering from respiratory diseases in addition to their cardiovascular disease(s) also had higher odds (2009: AOR = 1.45 [1.09–1.93]; 2014: AOR = 1.77 [1.31–2.39]; 2019: AOR = 1.54 [1.14–2.08]; merged sample: AOR = 1.56 [1.32–1.85]) of being immunized against influenza virus. Frequent meeting with a doctor or a specialist was considered a protective factor in terms of vaccination in 2009, in 2014, and in the merged database: those respondents who last visited their doctor more than a year ago had lower odds of being vaccinated against influenza (2009: AOR = 0.23 [0.09–0.59]; 2014: AOR = 0.39 [0.19–0.80]; merged sample: AOR = 0.40 [0.26–0.61]). However, it should be noted that the number of respondents was not distributed equally within all answer options regarding vaccination status, as is shown in Table 1. Therefore, there were scenarios in which less than 20 participants belonged to each category. Respondents in 2014 had significantly lower odds (AOR = 0.83 [0.70–0.97]) of proper immunization compared to participants in 2009. The same association was found for respondents in 2019, where participants had 32% lower odds (AOR = 0.68 [0.58–0.79]) of being immunized against the influenza virus compared to cardiovascular individuals in 2009.

## 4. Discussion

In this study, the EHIS database was used to conduct an analysis of influenza vaccine uptake and its determining factors. The analysis focused on the Hungarian population with self-reported cardiovascular diseases. We analyzed the temporal trend from 2009 to 2019, and, based on our findings, it can be stated that influenza vaccination coverage did not improve among patients with cardiovascular diseases over the years. In 2009, the vaccination coverage was 24%. This decreased to 22% by 2014, and, in the year 2019, it was merely 21%. All of this occurred even though free vaccinations are available for individuals suffering from cardiovascular diseases in Hungary [24].

In Hungary, the vaccination coverage among patients with cardiovascular diseases was notably low and fell significantly below the recommended 75% ratio set by the European Union [25]. The results of this study identified the factors influencing vaccination uptake. Older age was considered a protective factor among respondents in terms of vaccination as there was a significantly higher likelihood of receiving the influenza vaccination within this age group. This may be due to the fact that in Hungary, general practitioners monitor the elderly population, and vaccination is free of charge for this group. Furthermore, older patients with cardiovascular diseases might be prioritized by preventive public health interventions due to their increased risk, which could contribute to their higher odds of seeking influenza vaccination. This link has been confirmed by other studies [8,9,26]. One such study, published in 2020 in South Korea, found that the vaccination rate among patients with cardiovascular diseases under the age of 65 was lower than that of the older age group [27].

A higher level of education has proved to be a protective factor, and this assertion has been supported by the literature [5,28]. This is presumably due to the fact that a higher education level might be correlated with increased health awareness regarding the possible benefits of influenza vaccination. Additionally, respondents with a higher level of education may have better access to health and healthcare information, which could further facilitate their participation in preventive health initiatives. There was no significant correlation between sex and vaccine uptake in this study, although it was found that differences could be observed regarding sex in terms of vaccination strategies [29]. The likelihood of vaccine acceptance significantly increased among patients with cardiovascular diseases with poor self-perceived health status. Participants with poor self-perceived health status may be more inclined to prioritize immunization against influenza as a preventative approach for more protection against other comorbidities or health complications. Furthermore, this patient group may have more frequent meetings with their healthcare providers; therefore, the importance of influenza vaccination as part of their overall health management strategy could be emphasized during these encounters. These findings are supported by the literature, since a positive correlation was also found between poor self-rated health and the uptake of influenza vaccination, both in women and men [30]. The annual frequency of medical visits significantly influenced vaccination coverage. This result was consistent with the findings in the literature [31]. Frequent medical visits may increase the opportunities for healthcare providers to recommend and administer vaccinations, such as flu shots, thereby influencing vaccination coverage in a favorable manner among patients with cardiovascular diseases. Thus, less frequent encounters may contribute significantly to lower vaccination coverage. Smoking also proved to be a factor associated with vaccination coverage: non-smokers were more likely to be vaccinated [32,33]. Non-smokers could be considered as more health-conscious individuals who could be proactive about preventive measures to protect their health, such as getting vaccinated. Furthermore, in order to maintain the general health status of non-smokers, healthcare providers may grant priority to vaccination counseling and promotion for this patient group. One of the most important comorbidities are respiratory diseases; this further aggravates the fact that patients with cardiovascular diseases had a higher risk of severe outcomes. Therefore, comorbidities may play an important role in vaccine acceptance [34,35]. Patients with cardiovascular diseases who also have respiratory diseases are probably more aware of the risks associated with influenza and are probably more likely to accept vaccination as a preventive action to decrease these potential health risks.

### Strengths and Limitations

This study utilized robust datasets of the European Health Interview Surveys, providing representative samples of the Hungarian adult population. These surveys were supervised by Eurostat, ensuring a high quality of data collection, which could be considered as a potential strength of this study. Population health surveys are a particularly useful way to draw conclusions about the respondents’ health status, health behavior, and satisfaction with health services [36,37,38,39]. The analysis highlighted the longitudinal trends in influenza vaccination coverage among patients with cardiovascular diseases. Multiple logistic regression models enabled the identification of significant determinants of vaccination uptake, offering valuable insights for targeted intervention strategies. While the dataset offers comprehensive health-related information about the respondents, certain variables relevant to influenza vaccination behavior, such as attitudes and beliefs, were not measured, limiting the depth of this study’s analysis. Additionally, self-reported data may introduce bias or inaccuracies, particularly regarding vaccination status and health conditions. However, to address these potential limitations, several validation measures were implemented during the data collection process, such as a multistep stratified sampling strategy and the use of validated questionnaires. This study’s focus on Hungary may restrict generalizability to other regions with differing healthcare systems and cultural contexts.

## 5. Conclusions

Based on our results, it can be stated that influenza vaccination coverage was considerably low among individuals with cardiovascular diseases in Hungary. In addition, a decreasing trend was observed in vaccination coverage. The results also indicate that the likelihood of receiving an influenza vaccination among self-identified patients with cardiovascular diseases depends on several factors. These risk factors include young age, smoking, a low level of education, good self-reported health status, less frequent visits to health care personnel (such as general practitioners or specialists), and not reporting any respiratory diseases. It would be advisable to take these factors into account to develop effective and targeted vaccination strategies during the planning and implementation of vaccination campaigns, especially for those with cardiovascular diseases. Improving healthcare systems with a strong preventive approach while taking these factors into consideration could lead to better vaccination coverage. This could manifest, for example, in monitoring influenza vaccination among those at increased risk, such as patients with cardiovascular diseases, even at the primary care level, with the primary goal of reducing the spread and exacerbation of influenza among patients with cardiovascular diseases.

## Figures and Tables

**Table 1 vaccines-12-00360-t001:** Descriptive statistics by vaccination status of self-reported cardiovascular survey respondents. Significant findings (*p* < 0.05) are highlighted with “*”.

Factors		2009 N = 1997			2014 N = 2096			2019 N = 2179			Merged Sample N = 6272		
		Vaccinated within a Year N (%)	Not Vaccinated within a Year N (%)	*p*-Value	Vaccinated within a Year (N)	Not Vaccinated within a Year (N)	*p*-Value	Vaccinated within a Year (N)	Not Vaccinated within a Year (N)	*p*-Value	Vaccinated within a Year (N)	Not Vaccinated within a Year (N)	*p*-Value
Age group	18–64	174 (15%)	1022 (85%)	*p* < 0.001 *	162 (14%)	1028 (86%)	*p* < 0.001 *	110 (11%)	892 (89%)	*p* < 0.001 *	446 (13%)	2942 (87%)	*p* < 0.001 *
	65–X	305 (38%)	496 (62%)		305 (34%)	601 (66%)		347 (29%)	830 (71%)		957 (33%)	1927 (67%)	
Sex	Male	188 (23%)	625 (77%)	0.455	191 (21%)	700 (79%)	0.425	203 (22%)	724 (78%)	0.361	582 (22%)	2049 (78%)	0.688
	Female	291 (25%)	893 (75%)		276 (23%)	929 (77%)		254 (20%)	998 (80%)		821 (23%)	2820 (77%)	
Educational level	Primary	196 (27%)	536 (73%)	*p* < 0.001 *	27 (21%)	104 (79%)	0.007 *	119 (21%)	437 (79%)	*p* < 0.001 *	342 (24%)	1077 (76%)	*p* < 0.001 *
	Secondary	234 (23%)	803 (77%)		335 (21%)	1261 (79%)		226 (18%)	1012 (82%)		795 (21%)	3076 (79%)	
	Tertiary	49 (21%)	179 (79%)		105 (28%)	264 (72%)		112 (29%)	273 (71%)		266 (27%)	716 (73%)	
Type of residence	Urban	299 (24%)	941 (76%)	0.865	329 (23%)	1111 (77%)	0.356	325 (22%)	1158 (78%)	0.115	953 (23%)	3210 (77%)	0.163
	Rural	180 (24%)	577 (76%)		138 (21%)	518 (79%)		132 (19%)	564 (81%)		450 (21%)	1659 (79%)	
Marital status	Married or have a partner	256 (23%)	865 (77%)	0.174	273 (23%)	899 (77%)	0.209	269 (21%)	1034 (79%)	0.646	798 (22%)	2798 (73%)	0.695
	Not have a partner	223 (25%)	653 (75%)		194 (21%)	730 (79%)		188 (21%)	688 (79%)		605 (23%)	2071 (77%)	
Perceived income	Average/ Good	196 (22%)	682 (78%)	0.123	237 (24%)	762 (76%)	0.130	280 (23%)	930 (77%)	0.005 *	713 (23%)	2374 (77%)	0.173
	Bad	283 (25%)	836 (75%)		230 (21%)	867 (79%)		177 (18%)	792 (82%)		690 (22%)	2495 (78%)	
Self-perceived health status	Good	308 (22%)	1085 (78%)	0.003 *	327 (21%)	1247 (79%)	0.004 *	320 (19%)	1372 (81%)	*p* < 0.001 *	955 (21%)	3704 (80%)	*p* < 0.001 *
	Poor	171 (28%)	433 (72%)		140 (27%)	382 (73%)		137 (28%)	350 (72%)		448 (28%)	1165 (72%)	
BMI	Overweight/ obese	336 (23%)	1098 (77%)	0.354	342 (23%)	1154 (77%)	0.313	339 (21%)	1311 (79%)	0.387	1017 (22%)	3563 (78%)	0.608
	Not overweight/ obese	143 (25%)	420 (75%)		125 (21%)	475 (79%)		118 (22%)	411 (78%)		386 (23%)	1306 (77%)	
Smoker	Yes	68 (14%)	410 (86%)	*p* < 0.001 *	49 (11%)	394 (89%)	*p* < 0.001 *	61 (13%)	406 (87%)	*p* < 0.001 *	178 (13%)	1210 (87%)	*p* < 0.001 *
	No	411 (27%)	1108 (73%)		418 (25%)	1235 (75%)		396 (23%)	1316 (77%)		1225 (25%)	3659 (75%)	
Respiratory disease	Yes	96 (30%)	228 (70%)	0.009 *	85 (31%)	193 (69%)	*p* < 0.001 *	80 (28%)	208 (72%)	0.002 *	261 (29%)	629 (71%)	*p* < 0.001 *
	No	383 (23%)	1290 (77%)		382 (21%)	1436 (79%)		377 (20%)	1514 (80%)		1142 (21%)	4240 (79%)	
Last meeting with a doctor/or a specialist	<12 months	474 (25%)	1437 (75%)	*p* < 0.001 *	458 (23%)	1549 (77%)	0.005 *	446 (21%)	1636 (79%)	0.017 *	1378 (23%)	4622 (77%)	*p* < 0.001 *
	≥12 months	5 (6%)	81 (94%)		9 (10%)	80 (90%)		11 (11%)	86 (89%)		25 (9%)	247 (91%)	
Region (NUTS 2)	Central Hungary	106 (22%)	371 (78%)	0.196	139 (27%)	378 (73%)	0.003 *	137 (23%)	456 (77%)	0.016 *	382 (24%)	1205 (76%)	0.002 *
	Southern Great Plain	70 (22%)	243 (78%)		51 (16%)	259 (84%)		48 (17%)	228 (83%)		169 (19%)	730 (81%)	
	Southern Transdanubia	56 (28%)	146 (72%)		56 (26%)	163 (74%)		43 (19%)	180 (81%)		155 (24%)	489 (76%)	
	Northern Great Plain	78 (23%)	259 (77%)		65 (18%)	304 (82%)		59 (18%)	271 (82%)		202 (20%)	834 (81%)	
	Northern Hungary	81 (30%)	191 (70%)		62 (22%)	222 (78%)		57 (20%)	225 (80%)		200 (24%)	638 (76%)	
	Central Transdanubia	46 (22%)	160 (78%)		53 (25%)	155 (75%)		69 (29%)	171 (71%)		168 (26%)	486 (74%)	
	Western Transdanubia	42 (22%)	148 (78%)		41 (22%)	148 (78%)		44 (19%)	191 (81%)		127 (21%)	487 (79%)	

**Table 2 vaccines-12-00360-t002:** Analysis of the factors that were associated with influenza vaccination among self-reported cardiovascular respondents. Significant (*p* < 0.05) findings are highlighted with “*”.

Factors		2009 N = 1997			2014 N = 2096			2019 N = 2179			Merged Sample N = 6272		
		Adjusted Odds Ratio	95% Confidence Intervals		Adjusted Odds Ratio	95% Confidence Intervals		Adjusted Odds Ratio	95% Confidence Intervals		Adjusted Odds Ratio	95% Confidence Intervals	
Age group	65–X/18–64	3.58 *	2.80	4.57	3.08 *	2.43	3.89	3.17 *	2.47	4.07	3.22 *	2.80	3.70
Gender	Female/male	0.91	0.72	1.15	0.97	0.77	1.23	0.87	0.69	1.10	0.91	0.80	1.04
Educational level	Secondary/primary	1.29	0.99	1.68	1.59	0.99	2.57	0.96	0.72	1.28	1.14	0.96	1.35
	Tertiary/primary	1.08	0.71	1.64	2.21 *	1.29	3.78	1.74 *	1.21	2.52	1.57 *	1.25	1.97
Type of residence	Urban/rural	1.02	0.80	1.29	0.95	0.74	1.23	0.97	0.75	1.26	0.98	0.85	1.13
Marital status	Married or have a partner/not have a partner	1.00	0.78	1.27	1.18	0.92	1.50	1.16	0.91	1.48	1.11	0.97	1.27
Perceived income	Bad/good	0.96	0.75	1.22	0.83	0.65	1.07	0.87	0.68	1.11	0.89	0.78	1.02
Self-perceived health status	Poor/good	1.11	0.87	1.41	1.30 *	1.01	1.67	1.60 *	1.24	2.07	1.31 *	1.13	1.51
BMI	Not overweight or obese/overweight or obese	1.18	0.93	1.51	0.91	0.71	1.16	1.14	0.88	1.46	1.05	0.91	1.21
Smoker	No/yes	1.53 *	1.13	2.09	1.96 *	1.39	2.75	1.41 *	1.03	1.92	1.60 *	1.34	1.93
Respiratory disease	Yes/no	1.45 *	1.09	1.93	1.77 *	1.31	2.39	1.54 *	1.14	2.08	1.56 *	1.32	1.85
Last meeting with a doctor or a specialist	≥12 months <12 months	0.23 *	0.09	0.59	0.39 *	0.19	0.80	0.57	0.29	1.10	0.40 *	0.26	0.61
Region (NUTS 2)	Central Hungary	Reference											
	Southern Great Plain	1.11	0.77	1.61	0.64 *	0.44	0.94	0.83	0.56	1.22	0.83	0.67	1.03
	Southern Transdanubia	1.51 *	1.01	2.26	1.18	0.80	1.75	0.95	0.63	1.43	1.20	0.95	1.50
	Northern Great Plain	1.20	0.83	1.72	0.71	0.50	1.01	0.77	0.54	1.11	0.86	0.70	1.05
	Northern Hungary	1.71 *	1.18	2.47	0.97	0.67	1.42	0.96	0.66	1.40	1.16	0.94	1.44
	Central Transdanubia	1.05	0.69	1.59	1.04	0.70	1.54	1.54 *	1.07	2.22	1.20	0.96	1.51
	Western Transdanubia	0.98	0.63	1.51	0.83	0.55	1.27	0.81	0.54	1.22	0.86	0.67	1.09

## Data Availability

The data that support the findings of this study are available from the Hungarian Central Statistical Office, but restrictions apply to the availability of these data, which were used under license for this study; as a result, they are not publicly available. The data are, however, available from the corresponding author upon reasonable request and with prior permission from the Hungarian Central Statistical Office.

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
