# Peer review of "Influenza Vaccination Coverage among People with Self-Reported Cardiovascular Diseases—Findings from the Hungarian Implementation of the European Health Interview Survey"

_vaccines, 2024, doi:10.3390/vaccines12040360_

Round 1

Reviewer 1 Report

Comments and Suggestions for Authors

The manuscript (ID: vaccines-2914554) aimed to assess influenza vaccination coverage and its determinants among cardiovascular respondents from 2009 to 2019 based on data from the Hungarian implementation of the European Health Interview Survey.  

Comments:    

Abstract:

  • Lines 25-27: Add data on the connection between influenza vaccine uptake and smoking habits (non-smokers) and presence of respiratory disease among participants in this study.  

Introduction:

  • Line 69: Add a new paragraph in which the results of such studies from other countries that were conducted within the European Health Interview Survey (EHIS) studies of 2009, 2014, and 2019 should be described in more detail. 
  • Lines 70-73: Has the goal set in this paper been achieved, that is, `to assess and compare influenza vaccination coverage and its influencing factors among self-reported cardiovascular and non-cardiovascular respondents`? If not, correct either the objective of the work or present the results in this work in accordance with the defined objectives.   

Methods:

  • The section Methods substantially supplemented with data about `Study Settings`, `Study design`, `Study population`, `Study sample`, `Sample size calculation`, `Response ratr`, eligibility criteria, surveys, data collection, etc. State who collected the data in this study.    
  • Explain whether the presence of cardiovascular disease was the only indication for influenza vaccination.      
  • Lines 107-108: Does this paper present the above data on physical activity?     

Results:

  • Lines 237-242: Where are the listed results displayed?     
  • Lines 243-244: The title of Table 2 is missing.  

Discussion:

  • It is very concise, it is necessary to compare one's own results with the findings of similar studies in other countries.  
  • Provide possible explanations for significant independent associations of variables in this study, that is, between influenza vaccination uptake and age, education level, self-perceived health status, lower frequency of medical visits, smoking status, presence of respiratory disease.    

Reviewer 2 Report

Comments and Suggestions for Authors

The manuscript represents an analysis of information regarding influenza vaccination in Hungary. The manuscript may provide basic elements for health care policies. There are, however, some issues which were not very clear in the analysis. How was the data validated? The question refers in general to the criteria of influenza vaccination has changed in some countries between the first date up to date. What is the main reason for the resistance to vaccination? and why there is a high number of individuals who have not visited a doctor in more than 12 months? Interestingly, people with respiratory diseases had higher compliance for vaccination, but what respiratory diseases? The data on older individuals is interesting and should be compared with the younger ones. In addition, the data regarding overweight obese vs normal weight too. What is the incidence of influenza in Hungary on the different dates analysed? In the discussion, some proposals should be made. The conclusions are too long and should be concise. The study has limitations and should be stated. I am not sure about bias limitation depending on how the data was obtained.

Comments on the Quality of English Language

Some grammatical mistakes were encountered.

Round 2

Reviewer 1 Report

Comments and Suggestions for Authors

Thank you for the opportunity to re-review this paper. The authors have responded to all my comments. In the context of some of my comments, the authors responded correctly and made appropriate changes in the revised paper (in the Abstract, Introduction, and, especially, in the Discussion section). On the other hand, they gave satisfactory explanations for several of my comments and made such corrections in this paper. Overall, the corrections made have improved the clarity and informativeness of this paper. Thanks to the authors.      

Author Response

Reviewer 1

Thank you for the opportunity to re-review this paper. The authors have responded to all my comments. In the context of some of my comments, the authors responded correctly and made appropriate changes in the revised paper (in the Abstract, Introduction, and, especially, in the Discussion section). On the other hand, they gave satisfactory explanations for several of my comments and made such corrections in this paper. Overall, the corrections made have improved the clarity and informativeness of this paper. Thanks to the authors.    

Dear Reviewer, we would like to thank you for all the inputs you gave which considerably improved the overall manuscript. As requested by the other Reviewer, we have made some alterations in the Discussion and Conclusions sections. Also, the overall English was improved by us and the English Language Editing Services of the MDPI.

Reviewer 2 Report

Comments and Suggestions for Authors

The manuscript was improved by adding new requested data. There have been some changes in the text. I do not agree with the authors concerning epidemiological data from Hungary, in particular the incidence of infection, and the lack of a section on the limitations of the study. However, I consider that the main points were covered.

Comments on the Quality of English Language

Several grammatical mistakes were encountered.

Author Response

Reviewer 2

The manuscript was improved by adding new requested data. There have been some changes in the text. I do not agree with the authors concerning epidemiological data from Hungary, in particular the incidence of infection, and the lack of a section on the limitations of the study. However, I consider that the main points were covered.

We would like to express our thanks to the Reviewer for the further comments and suggestions in order to improve the quality of the manuscript.

It is relatively difficult to accurately assess data regarding influenza vaccination coverage and incidence of the disease per influenza season in Hungary, as the officially available data for Hungary are unfortunately not completely unified in terms of data availability and presentation. Previously, several institutes and centres were responsible for data ownership and therefore significant changes had been made to organisational structures. These changes led to source data migration in many cases, respectively. At present, in Hungary, a sentinel surveillance service (https://www.nnk.gov.hu/index.php/jarvanyugyi-es-infekciokontroll-foosztaly/leguti-figyeloszolgalat.html) provides data on influenza for the number of new cases between week 40 and week 20, broken down by week, but unfortunately a clear summary report is not available regarding the total incidence of influenza cases. However, the Hungarian Central Statistical Office (https://www.ksh.hu/stadat_files/ege/hu/ege0028.html) provides limited access to the overall incidence data for Hungary regarding reported influenza for the recent years. The estimated case numbers were the following: 2008=231,000, 2009=323,000, 2013=198,000, 2014=141,000, 2018=433,000, 2019=389,000. These data suggest that the epidemiological potential for influenza infection has been higher in recent years. But we would like to respectfully note that as the infection usually occurs at the end of the year as well as at the beginning of each year, we do not fully agree with the interpretation of the number of new cases for the whole calendar year, as it may be misleading. Despite this, surveillance data from the Annual Epidemiological Report of ECDC also shows data for Hungary; unfortunately, the way these are reported is not completely consistent for all years analysed during our study, which is why we have not reported these data in the manuscript.

Furthermore, we would like to respectfully note that after some deliberation, the research team decided not to report data regarding the incidence of the disease in the manuscript, because the main focus of the study was on the coverage of vaccination among cardiovascular patients and the factors that influence vaccination coverage. In our opinion, it would be appropriate from a public health perspective to achieve the recommended 75% vaccination coverage rate in all study years among this high-risk group. We certainly agree with the Reviewer that knowing this data and examining this association with vaccination coverage could be of great interest for both experts and policy makers, but unfortunately due to the focus of this study this was not done due to reasons beyond our control.

Subsequently, we respectfully thank you for your comments on the language of the manuscript. The English text of the whole paper was double-checked by the Authors. In addition, we have also asked the English Language Editing Services of the MDPI to edit out manuscript with track changes.

Finally, we would like to apologize for not correctly implementing the original request regarding the limitations of the manuscript. As requested by the Reviewer, the clarification of the strength and limitations section has been done which we hope is in line with the original comment made by the Reviewer.

Now thy can be read: “This study utilized robust datasets of the European Health Interview Surveys, providing representative samples of the Hungarian adult population. These surveys were supervised by Eurostat, ensuring a high quality of data collection, which could be considered as a potential strength of the present study. (…) Additionally, self-reported data may introduce bias or inaccuracies, particularly regarding vaccination status and health conditions. However, to address these potential limitations, several validation measures were implemented during the data collection process, such as a multistep stratified sampling strategy and the use of validated questionnaires.”